# Automated Accelerometer-Based Gait Event Detection During Multiple Running Conditions

**DOI:** 10.3390/s19071483

**Published:** 2019-03-27

**Authors:** Lauren C. Benson, Christian A. Clermont, Ricky Watari, Tessa Exley, Reed Ferber

**Affiliations:** 1Faculty of Kinesiology, University of Calgary, Calgary, AB T2N 1N4, Canada; christian.clermont@ucalgary.ca (C.A.C.); rickywatari@usp.br (R.W.); tessa.exley@ucalgary.ca (T.E.); rferber@ucalgary.ca (R.F.); 2Faculty of Nursing and Cumming School of Medicine, University of Calgary, Calgary, AB T2N 1N4, Canada; 3Running Injury Clinic, University of Calgary, Calgary, AB T2N 1N4, Canada

**Keywords:** wearable technology, gait event detection, running, accelerometer, initial contact, toe off

## Abstract

The identification of the initial contact (IC) and toe off (TO) events are crucial components of running gait analyses. To evaluate running gait in real-world settings, robust gait event detection algorithms that are based on signals from wearable sensors are needed. In this study, algorithms for identifying gait events were developed for accelerometers that were placed on the foot and low back and validated against a gold standard force plate gait event detection method. These algorithms were automated to enable the processing of large quantities of data by accommodating variability in running patterns. An evaluation of the accuracy of the algorithms was done by comparing the magnitude and variability of the difference between the back and foot methods in different running conditions, including different speeds, foot strike patterns, and outdoor running surfaces. The results show the magnitude and variability of the back-foot difference was consistent across running conditions, suggesting that the gait event detection algorithms can be used in a variety of settings. As wearable technology allows for running gait analyses to move outside of the laboratory, the use of automated accelerometer-based gait event detection methods may be helpful in the real-time evaluation of running patterns in real world conditions.

## 1. Introduction

Running gait analyses can be used for injury prevention and treatment, as well as in performance enhancement [1]. With their portability and capability to record, store, and stream movement data, accelerometers are well-suited for gait analysis during prolonged runs in a runner’s natural environment [2,3,4,5]. Typically, the variables of interest that are used for running gait analyses are basic spatiotemporal gait parameters or more advanced features that are related to movement quality determined for each stance phase, step, or stride, and these variables require the identification of initial contact (IC) and toe off (TO) events for each running step [6,7]. Access to these variables allows for the clinician, coach, or runner to obtain fast and accurate feedback about running patterns [3], and they may be used in gait retraining programs to prevent injury [8]. However, gait event detection methods have been historically developed using three-dimensional (3D) motion capture and force plate data [9,10,11,12], and these methods are limited to indoor running facilities and are only possible using expensive equipment. Therefore, automatic accelerometer-based methods of detecting running gait events are crucial for real-world running gait analyses.

Patterns in accelerometer signals have been used for the identification of gait events during running. In particular, gait event detection methods have been developed using accelerometers that are placed on either the foot [13,14,15] or the back [16,17,18,19]. However, these algorithms have been defined or described with varying levels of detail, from a basic description of the shape of the accelerometer waveform [16] to step-by-step instructions for identifying gait events [20]. Additionally, various degrees of rigor have been used to validate accelerometer-based gait event detection methods. In some cases, there was no validation protocol [13], or the accelerometer signal was visually inspected to identify the gait events [14]. Other studies compared the locations of corresponding gait events between the accelerometer-based analysis and concurrent video [16,18,20], insole pressure [17], or kinematic analyses [15,19]. In contrast, the use of ground reaction force thresholds from a force plate is considered as the gold standard for identifying gait events, which Mo and Chow [21] used as a point of comparison for gait events that were determined from accelerometers placed on the back and foot. However, the validation protocols that were used to develop the aforementioned methods for gait event detection have contributed to a lack of robustness necessary to account for varying intrinsic and extrinsic conditions encountered by runners in real-world conditions.

A significant flaw in the accelerometer-based methods for gait event detection is that the algorithms are based on the assumption of a consistent running pattern, irrespective of running conditions or subject-specific running gait patterns [22]. For example, absolute thresholds (e.g., 2 g or 8 g) have been included in several methods, with no consideration for when the accelerometer signal does not achieve the required threshold [14,15]. Likewise, peak detection is generally a key component of these algorithms, but again no contingencies are offered for when a peak does not exist, occurs earlier or later than expected, or when multiple peaks are present [19]. A possible reason for these oversights is that the published methods have been established in controlled settings (e.g., treadmill, track, or indoor laboratory space), with small sample sizes of distinct subgroups of runners (e.g., competitive only or recreational only), and/or using a limited number of steps to test the algorithm [13,14,15,16,17,18,19,20]. The automation of robust gait event detection methods would enable the evaluation of many steps across multiple running conditions.

Therefore, this study provides detailed gait event detection algorithms for running that were modified from previously-described patterns of foot and back accelerometer signals [14,19], and were optimized for automatic gait event detection. These algorithms were validated during running on an instrumented treadmill, and the differences in magnitude and variability of the back and foot gait event detection methods were examined during various running conditions, including different speeds, foot strike patterns, and outdoor running surfaces. The goal of this paper is to provide freely-available automated gait event detection algorithms for running that can be used in future research to further our understanding of real-world running patterns, leading to improvements in performance and a reduction in running-related injuries.

## 2. Materials and Methods

This study was carried out in accordance with the recommendations of the University of Calgary research ethics board (REB16-1183 approved 6 July 2016; E-24519 approved 3 May 2012) with written and informed consent being obtained from all participants. Data were collected in three separate experiments, using different sets of participants and running conditions for each experiment. The participants were adult male and female recreational runners with no running related injuries in the previous three months. In experiment 1, participants ran on an instrumented treadmill at different speeds. In the second experiment, a new set of participants ran on an indoor track surface at different speeds and with different foot strike patterns. Finally, for experiment 3, a third set of participants ran outdoors on both the sidewalk and grass surfaces on multiple days. In all cases, two tri-axial accelerometers (Shimmer3^®^ range: +/− 16 g, sampling rate: 201.03 Hz; Shimmer Inc., Dublin, IE) were secured to the dorsum of the right foot and the lower back near the center of mass. For the foot, the X, Y, and Z axes were oriented in the medial-lateral (ML; + to the right), anterior-posterior (AP; + to the posterior), and vertical (VT; + superior) directions, respectively. For the back, the X, Y, and Z axes were oriented in the ML (+ to the right), VT (+ superior), and AP (+ to the posterior) directions, respectively.

### 2.1. Data Collection

#### 2.1.1. Experiment 1: Instrumented Treadmill

Twelve healthy recreational runners (8M, 4F; age = 26.2 (3.8) years, height = 1.78 (0.06) m, and mass = 71.5 (7.1) kg) wore their preferred clothing and the same standardized running shoe (Nike Air Pegasus) for the experiment. Following 2–5 min of warm-up and acclimation to the instrumented, split-belt treadmill (Bertec Inc. sampling rate: 1000 Hz, Columbus, OH), the participants ran at slow, intermediate, and fast speeds (2.7 m/s, 3.3 m/s, and 3.6 m/s) [23] for 90 s, while data were recorded using the foot and back-mounted tri-axial accelerometers, as well as the force platform. Immediately before and after each speed condition trial, the participants were instructed to jump three times to synchronize the accelerometers and force plate data for analysis [24].

#### 2.1.2. Experiment 2: Indoor Track

Twenty healthy recreational runners (10M, 10F; age = 27.1 (3.4) years; height = 1.78 (0.06) m; mass = 72.2 (7.2) kg) wore their preferred clothing and shoes. The participants first ran a 200 m (one lap) warm up, followed by two 60-m trials along a straightaway portion of the track at their preferred speed and foot strike pattern. Timing gates were placed at each end of the straightaway to record the time to run 60 m and the two-trial average was used to quantify the preferred speed. Participants then ran two 60-m trials under each of three speed conditions (preferred, 25% faster than preferred, and 25% slower than preferred) [7,19] and two different foot strike conditions (rearfoot strike and forefoot strike) for a total of 12 trials. If a trial fell outside of ±5% of the required speed, then it was rejected and repeated before advancing to the next trial [25,26]. All trials of a single foot strike pattern were completed before running with the second foot strike pattern, and both of the trials of the same speed and foot strike pattern were completed before changing speeds. However, the order of foot strike patterns and the order of speed conditions was randomized for each participant. At least 30 s were provided for recovery, and verbal feedback regarding speed was provided between each trial.

#### 2.1.3. Experiment 3: Outdoor

Twenty-two healthy runners (11M, 11F; age = 37.6 (11.0) years; height = 1.72 (0.09) m; mass = 70.1 (10.5) kg) wore their preferred clothing and shoes. First, the runners completed a 5–10 min warmup at their own pace. Next, the data were recorded as the participants ran on a continuous sidewalk surface at their preferred running speed for approximately 300 m, stopped, turned around, and ran approximately 300 m back to the starting point. They then ran the same 600 m (2 × 300 m) route on a continuous grass surface next to the sidewalk at their preferred speed. This process was repeated for a total of 16 trials of 300 m, for a total distance of 4800 m. The order of the surface conditions was randomized, such that some of the participants ran on grass first and others ran on sidewalk first. Participants also returned on a separate day to complete the same running course, resulting in nearly 10 km of data to analyze for each participant.

### 2.2. Data Processing

All data processing was done using custom MATLAB software (v9.5.0.1033004 (R2018b) Update 2, Mathworks, Inc., Natick, MA, USA). The back accelerometer signal was aligned with gravity [27] and rotated such that the positive ML axis pointed right, the positive AP axis pointed anteriorly, and positive VT axis pointed superiorly [28]. No alignment was performed for the foot accelerometer signal [14], therefore the foot VT axis generally pointed superiorly during standing. The foot and back accelerometer and force plate signals were preprocessed using a fourth order zero-lag Butterworth filter, with cutoff at 10 Hz and no normalization of the signals [29].

To ensure that the participants achieved a steady-state running pattern for the data that were included in the analysis, and to remove any potential gait irregularities that are associated with anticipating the termination of the trial, the beginning and end of each trial was trimmed. Specifically, for the treadmill running, 15 s from each end was trimmed leaving 60 s of data per trial. For the indoor track and outdoor conditions, the first and last 5% of each trial was removed. Since the participants ran at preferred speed for the indoor track and outdoor conditions, but a set speed for the treadmill condition, the amount that was trimmed from the indoor track and outdoor conditions was a percent of the data collected, while a set amount of time was trimmed from all treadmill trials. Furthermore, to avoid any potential compensations for adjustments to the mechanical acceleration of the treadmill and the set running speed, a greater percentage of data was trimmed from the treadmill trials than the overground indoor track and outdoor trials.

### 2.3. Algorithm Description

The accelerometer-based gait event detection algorithms and the force plate-based gold standard methods were developed based on several assumptions. First, it was assumed that the time between contralateral steps would be between 0.25 s and 0.50 s, and twice those values for the ipsilateral steps. These constraints correspond to a running cadence of between 240 steps/min and 120 steps/min, respectively, which encompasses the reported preferred running cadences of between 150–192 steps/min [30,31,32,33], with the added advantage that this assumption allows ample room to accommodate for a far wider range of running cadences. Additionally, it was assumed that TO would occur no earlier than 0.1 s following IC [34]. The gold standard for determining IC and TO events were the 10 N and 25 N thresholds, respectively, in the magnitude of the vertical ground reaction force [21]. Furthermore, the accelerometer-based algorithms were developed according to the specific orientation assumptions that are described in Section 2.2. The previously described accelerometer patterns by Strohrmann et al. [14] and Lee et al. [19] were chosen as the basis of the foot and back algorithms in the current study, as they relate to specific biomechanical features of running steps (e.g., peak accelerations and decelerations in three dimensions) and they have been used in subsequent studies to identify gait events [21], however, some modifications were introduced to avoid the utilization of acceleration thresholds of a specific magnitude and to accommodate the broader variability in observed running patterns. Detailed steps of the proposed algorithm for identifying IC and TO for a segment of running data are illustrated in Figure 1 and Figure 2.

### 2.4. Algorithm Implementation

The gait event detection algorithms were implemented with the goal of application for real-time gait analysis. A sliding window approach was used to mimic the requirement of processing smaller sections of incoming data, rather than a complete trial at once [7]. Additionally, due to the infinite number of potential starting points for a window relative to the gait cycle, it was possible that some of the gait events would be improperly identified due to their position within the window if they were evaluated only once. Thus, overlapping windows were used to increase the possibility that all potential gait events would be identified. The sliding window approach required the specification of a window size and magnitude of change in window start time, with the number of frames of the shift dependent on the sampling rate of the data [35]. For a given window size, the first window of data was processed according to the algorithms that are defined in Section 2.5, and the IC and TO events identified in that window of data were retained as potential gait events. The window was then advanced by a given change in window start time and the potential IC and TO events were identified for the new window. Only unique potential gait events were retained.

Once the potential gait events were passed by the sliding windows, they were evaluated to create a final list of IC and TO events for the trial. In this evaluation process, three consecutive potential IC events were analyzed first. If consecutive potential IC events were too close to each other, according to the cadence assumptions that are defined in Section 2.5 (less than 0.25 s between contralateral steps for the back accelerometer and less than 0.5 s between ipsilateral steps for the foot accelerometer), the next consideration was to determine whether both IC events were within a reasonable range of the previously established IC and the next potential IC. If both IC events were viable, according to the cadence assumptions, the potential IC with the greatest magnitude of the accelerometer signal (resultant for foot, VT axis for back) at the potential IC point was retained.

For each retained IC, the potential TO events were considered next according to the following procedure. Any TO that occurred prior to or at the same time as the retained IC was removed. If two consecutive TO events were between the retained IC and the next IC, or if the two consecutive TO events were too close to each other but within a reasonable range of the established IC, according to the cadence assumptions, the potential TO with the greatest magnitude of the accelerometer signal (in the negative VT axis for both the foot and back) at the potential TO point was retained. Once an IC and a corresponding TO were established, the step side (right or left) was only determined for the back accelerometer (see Figure 2), which was confirmed based on the fact that the foot accelerometer was only placed on the right foot. The MATLAB code for these automated algorithms, including a real-time visualization, is publicly available (see Appendix A).

The computational performance of the automated algorithms was quantified by calculating the output rate for each algorithm. Specifically, for each frame of data in the trial, a timer was started the first time that a frame appeared in one of the sliding windows while using the *tic* function in MATLAB. Once the frame was passed by the sliding windows and all potential gait events up to and including that frame were evaluated, the frame timer was stopped using the *toc* function in MATLAB. The output rate for each frame was calculated as one divided by the time to process the frame, and the mean output rate for all frames in the trial represented the output rate for the algorithm. To facilitate comparisons regarding the output rate for other computers, the total execution time on the computer used in this analysis for all six tasks of the MATLAB benchmark utility *bench*, averaged over 10 executions, was 2.5060 s.

To check the veracity of the retained gait events and the determined step side for the back algorithm, the IC and TO events were compared across the foot, back, and gold standard force methods. If a step was identified in one method without a corresponding step in another method, it was recorded as a skipped step for the deficient method. Since all of the steps recorded using the foot method were right-sided steps, the number of incorrectly determined step sides by the back accelerometer method was established using the back-foot comparison. Additionally, while using the foot-force and back-force comparisons, the mean time difference for IC, TO, and ground contact time (GCT; the time the foot is on the ground or TO-IC) was determined as foot minus force (foot-force) and back minus force (back-force).

As the most neutral and controlled condition in this study, the intermediate (3.3 m/s) speed condition from the instrumented treadmill trials was used to identify the optimal sampling rate, window size, and the change in window start time parameters for the automated algorithms. The force, foot, and back data from these trials were resampled and gait events were identified using all combinations of three sampling rates (50 Hz, 100 Hz, 200 Hz), three window sizes (1 s, 2 s, 5 s), and six changes in window start times (0.005 s, 0.01 s, 0.02 s, 0.1 s, 0.5 s, 1 s). The optimal parameters were determined based on four criteria: high output rate, minimal skipped steps, minimal wrong step sides that were identified using the back algorithm, and minimal differences in IC, TO, and GCT for the foot-force and back-force comparisons. Using those optimal parameters, the differences and limits of agreement (LoA) in IC, TO, and GCT for the foot-force and back-force comparisons were reported while using a Bland–Altman plot.

### 2.5. Algorithm Testing

Using the sampling rate, window size, and change in window start time parameters that were established in Section 2.4, gait events were identified from the foot and back accelerometer signals for all trials of all running conditions described in Section 2.1. The time difference between the back and foot methods (back minus foot; back-foot) was determined for each IC and TO, and the mean and standard deviation of the back-foot difference was calculated for each participant and each trial type. For example, for the indoor track condition, the arrays of the back-foot differences for IC and TO were combined for both trials of a given speed and foot strike pattern condition, and then the mean and standard deviation were calculated. Likewise, all the trials for outdoor running on the same surface were combined prior to finding the mean and standard deviation of the back-foot difference for IC and TO.

The mean and standard deviation of the back-foot difference were the dependent variables in assessments regarding the effect of the different running conditions and gait event types on the difference between the back and foot methods. Using the data from the instrumented treadmill trials, the effect of speed and gait event type was determined with two separate 3 × 2 repeated measures ANOVAs; one for the mean and one for the standard deviation of the back-foot difference, with speed factors of 2.7 m/s, 3.3 m/s, and 3.6 m/s, and IC and TO as the gait event factors. Next, the data from the indoor track trials was used to determine the effect of speed, foot strike pattern, and gait event type with two additional 3 × 2 × 2 repeated measures ANOVA, with speed factors of preferred, fast and slow, foot strike pattern factors of rearfoot and forefoot, and IC and TO as comprising gait event factors. Finally, using the data from the outdoor trials, the effect of surface and gait event type was determined with two separate 2 × 2 repeated measures ANOVAs, with sidewalk and grass as the surface factors, and IC and TO as the gait event factors. The significance for all ANOVAs was determined at *p* < 0.05, and a Bonferroni adjustment was used to determine statistical significance for all follow up tests. All of the statistical analyses were performed in SPSS (v25, SPSS, Inc., Chicago, IL).

## 3. Results

### 3.1. Algorithm Implementation

Differences were apparent in the output rate between the force, foot, and back methods when the change in window start time was large (≥0.1 s), with larger differences for smaller window sizes (Figure 3). Sliding windows consisting of smaller window sizes with larger changes in the window start time (e.g., window size = 1 s, change in window start time = 1 s) were processed fastest and they had the highest output rate. In these conditions, and with the computer that was used in this study, the force data could be processed at a rate of 250–460 Hz for sampling rates of 200–50 Hz, respectively. Under these same conditions, the foot data had an output rate of 150–210 Hz, while the back data had an output rate of 90–100 Hz. With the assumption that a step takes a maximum of 0.5 s, at a sampling rate of 200 Hz, one step could be processed in 0.67 s for the foot, and 1.1 s for the back. However, in real time, the processing speed would be moderated by the sampling rate and on-board capabilities of the device itself. Furthermore, for larger window sizes and/or a smaller change in window start time, the output rate decreased, and the lowest output rate for all of the methods was about 2 Hz, regardless of window size and with the overlapping window shifting one frame at a time (0.005 s) for data that were sampled at 200 Hz. Using these parameters, it would take 50 s to process a step of 0.5 s sampled at 200 Hz.

Across all parameter combinations, the average number of the recorded steps was 1942 (range: 1847–1950). The number of skipped steps was reported as a total of all steps from all participants. Under most sampling rates and window size and change in window start time parameters, there were zero skipped steps using the force, foot, and back methods (Figure 4). However, the number of skipped steps was above zero when the sliding windows did not overlap (i.e., window size = 1 s, change in window start time = 1 s), with the largest number of skipped steps for the foot. In the foot-force comparison, there was also at least one step that was skipped by the foot method for all sampling rates and window sizes when the change in window start time was 1 s.

While using the back-foot comparison, the total number of steps from all participants with a left or right side identified incorrectly via the back algorithm was consistent across all sampling rate, window size, and change in window start time parameters (Figure 5). Out of an average of 1942 recorded steps, between 322–377 steps were assigned to the wrong side. There were fewer incorrect step sides assigned when the window size was 1 s and the change in window start time was 1 s, which also corresponded to conditions that produced a higher rate of skipped steps.

The gait events that were identified using the foot and back algorithms were compared to the gold standard force plate method, and the mean difference in timing for IC, TO, and GCT for foot-force and back-force were reported for the different sampling rate, window size, and change in window start time parameters (Figure 6). For both foot-force and back-force comparisons, the mean difference for IC was consistent across all parameter values, with foot IC being identified approximately 0.015 s prior to force IC (i.e., negative foot-force difference) and back IC identified approximately 0.05 s after force IC (i.e., positive back-force difference). For both foot-force and back-force comparisons, the accelerometer-based TO was about 0.02–0.03 s after the force TO, but the magnitude of the mean difference tended to increase for smaller values of the change in window start time parameter in the foot-force comparison. As GCT is equal to TO-IC, the mean difference in GCT follows the previously mentioned patterns, such that there was a smaller magnitude of difference in the foot-force comparison for smaller values of the change in the window start time parameter and the difference was positive or negative for the foot-force or back-force comparisons, respectively.

Overall, the accuracy (skipped steps, wrong step side assignment, magnitude of difference from gold standard) of gait event detection improved with small values for the change in window start time, but the output rate also decreased under those parameter conditions. Since the accuracy tended to converge for the three smallest values for the change in window start time, and little, if any, differences were observed for changes in the sampling rate and window size, the following parameters were chosen for all subsequent analyses: sampling rate = 200 Hz, window size = 2 s, and change in window start time = 0.02 s.

A Bland–Altman plot indicated the same mean offset, as demonstrated in Figure 6, and the 95% LoA between foot-force and back-force for IC, TO, and GCT (Figure 7). The range of the 95% LoA were smaller for back-force than foot-force for all events. The wider LoA for TO than IC for the foot-force comparisons were mainly due to two participants (blue and green dots in Figure 7c), with a consistent offset where foot TO was recorded approximately 0.15 s after the force TO.

### 3.2. Algorithm Testing

The mean and standard deviation of the difference between the back and foot algorithms for each event (IC and TO) and running condition (e.g., speed, foot strike pattern, surface) are reported for the instrumented treadmill, indoor track, and outdoor data collections (Table 1). In all conditions, IC that was detected with the back method occurred later than the IC detected with the foot method (positive mean back-foot difference), while TO that was detected with the back method occurred prior to TO that was detected with the foot method (negative mean back-foot difference).

During the treadmill trials, there was no speed x event interaction for the mean or standard deviation of the difference between the back and foot algorithms, and no main effect of speed on the mean back-foot difference (*p* > 0.05). However, there was a main effect of speed (F(1.106, 12.161) = 5.066, *p* = 0.041) on back-foot variability, but none of the follow up comparisons were significant at the Bonferroni-adjusted α = 0.017. There was also a main effect of event on the mean (F(1, 11) = 40.984, *p* < 0.001) and standard deviation (F(1, 11) = 12.361, *p* = 0.005) of the back-foot difference, with a greater mean difference for IC (0.067 (0.014) s) than TO (−0.011 (0.045) s), and greater variability in the back-foot difference for TO (0.030 (0.017) s) than IC (0.016 (0.007) s).

For the indoor track trials, the only significant interaction for the mean difference between the back and foot algorithms was the foot strike x event interaction (F(1, 19) = 11.840, *p* = 0.003). There was a simple main effect of event with greater back-foot difference for IC than TO in both rearfoot and forefoot strike running, and a simple main effect of foot strike pattern with greater back-foot difference for rearfoot than forefoot in both IC and TO (Figure 8). There was no significant main effect of speed (*p* > 0.05). The only significant interaction for the standard deviation of the back-foot difference was the speed x event interaction (F(1.427, 27.104) = 6.787, *p* = 0.008). The simple main effect of event revealed greater variability in back-foot difference for TO than IC at every speed, and the simple main effect of speed showed greater variability in back-foot difference at fast when compared to the preferred speed for only IC (Figure 9). There was also a main effect of foot strike pattern on the standard deviation of the back-foot difference (F(1, 19) = 14.383, *p* = 0.001), with greater variability for the rearfoot strike pattern (0.060 (0.018) s) than for the forefoot strike pattern (0.049 (0.009) s).

In the outdoor trials, there was a significant surface x event interaction for the mean difference between the back and the foot algorithms (F(1.427, 27.104) = 6.787, *p* = 0.008). There was also a simple main effect of event with greater back-foot difference for IC than TO in both sidewalk and grass running, and a simple main effect of surface revealed a greater back-foot difference for sidewalk than grass for TO, but not IC (Figure 10). There was no surface x event interaction for the standard deviation of the difference between the back and foot algorithms (*p* > 0.05). There was a main effect of event on the standard deviation of the back-foot difference (F(1, 21) = 84.627, *p* < 0.001), with greater variability in the back-foot difference for TO (0.065 (0.019) s) than IC (0.032 (0.019) s).

## 4. Discussion

The purpose of this study was to provide detailed and automated gait event detection algorithms for running, while using an accelerometer that was placed on either the foot or back and across a variety of running conditions. While these algorithms were developed using patterns that were described by Lee et al. [19] and Strohrmann et al. [14], modifications were made to enable the algorithm to handle variations in running patterns, as is the case when processing data from multiple runners, in multiple conditions, and over many steps. The automated algorithms were validated against a gold standard force plate method of identifying IC and TO. Since the force plate method could not be used in the non-treadmill running conditions in this study, an evaluation of the accelerometer-based gait event detection algorithms was achieved by examining the mean and variability of the difference between the back and foot methods in different running conditions.

For most of the participants, the back algorithm detected IC and TO after the force plate method and the foot algorithm detected IC and TO prior to the force method. Additionally, the 95% LoA were narrower for the back algorithm than the foot algorithm, and the 95% LoA were wider for TO than IC across both methods. In a previous study, Mo and Chow reported that the back accelerometer method that was developed by Lee et al. [19] can identify IC and TO during running, with a mean absolute difference from the reference force plate of 0.006 ± 0.005 s and 0.020 ± 0.008 s, respectively, while the foot accelerometer method that was developed by Strohrmann et al. [14] performs better at IC (0.004 ± 0.005 s) but worse at TO (0.028 ± 0.008 s) [21]. These previously reported values [21] are lower than the values of the current study for IC, while the mean offset values for TO are similar. Therefore, there appears to be greater variability in our results, as evidenced by the range of the 95% LoA (foot-force: IC = 0.084 s, TO = 0.232 s; back-force: IC = 0.058 s, TO = 0.078 s). The greater variability may be due to the utilization of 1954 steps, as opposed to just 10 trials of one step from 11 recreational runners across two speed conditions (220 total steps) that were analyzed by Mo and Chow [21]. Furthermore, the between-subject variability appears to be greater than the within-subject variability, as can be observed in Figure 7. Based on this observation, it is possible that the protocol that was used to sync the force plate and accelerometer signals using jumps at the beginning and end of each trial introduced individual offsets between the force plate and accelerometer signals that carried over into the validation analysis. Therefore, future research involving different synchronization methods may be necessary to help answer this question and possibly improve upon the provided algorithms.

Based on the magnitude and direction of the IC offset for the foot (negative) and back (positive) as compared to the force method, it was expected that the back-foot difference for IC would be positive. Since the TO offset was close to zero for both the foot-force and back-force comparisons, it was expected that the back-foot difference for TO would be less than the back-foot difference for IC. This was observed in all running conditions that were examined in this study. Additionally, greater variability was observed for TO than IC in all cases, which was consistent with the wider 95% LoA for TO than IC for the foot-force and back-force comparisons. Thus, the effect of event on the magnitude and variability of the back-foot difference was similar across running conditions, which can be considered to be a strength of the current study.

Another significant strength of the current study was that the effect of speed, foot strike pattern, and surface on the magnitude and variability of the back-foot difference was also examined. Speed only influenced the variability of the back-foot difference in some cases, and while there was a main effect of speed on back-foot difference variability for treadmill running, the individual comparisons of speed conditions were not significantly different. Moreover, during the indoor track trials, there was only greater variability of the back-foot difference for the fast condition as compared to preferred speed running and only for IC. As well, different foot strike patterns appeared to shift the relative timing of gait events for the foot and back methods without changing the relationship between IC and TO. For example, a forefoot strike pattern, when compared to a rearfoot strike pattern, resulted in smaller (less positive) back-foot difference for IC, while the magnitude of the back-foot difference for TO was greater (more negative). There was also greater variability in the back-foot difference for rearfoot as compared to forefoot strike running. Finally, the only effect of surface was a greater (more negative) back-foot difference for TO in the grass condition when compared to the sidewalk condition.

In all situations where there was an effect of running condition on the magnitude or variability of the back-foot difference, the observed changes could be either due to runners adopting a different running pattern that influenced the accelerometer signal at the foot and/or back or a failure of either algorithm to detect the proper gait events. However, the ability to validate the back and foot algorithms against a gold standard in non-force plate environments is not possible. Thus, based on the consistent effect of event on the magnitude and variability of the back-foot difference across the running conditions, it is likely that the observed effects of running conditions were due to changing running patterns. Future research is necessary to help confirm, or refute, this hypothesis. Additionally, future studies may consider the further validation of these algorithms with a larger sample of participants.

In conclusion, this study provides automated algorithms for identifying running gait events from accelerometers that were placed on the back or foot. Through a process of systematic validation against a gold standard force plate gait event detection method, and comparisons between the back and foot, these algorithms can be used in a variety of running conditions. As wearable technology allows for running gait analyses to move outside of the laboratory setting, the use of automated accelerometer-based gait event detection methods may be helpful in evaluating running patterns in the real world and these algorithms are freely available for the research community as well as wearable technology companies.

## Figures and Tables

**Figure 1 sensors-19-01483-f001:**
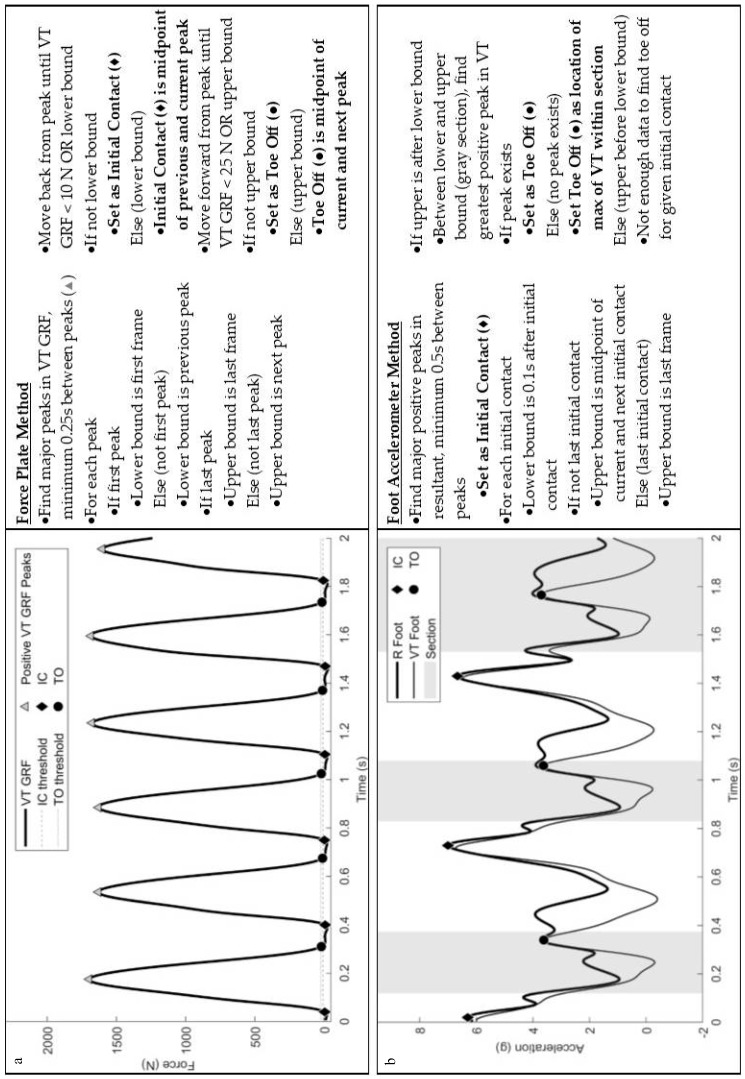
Illustration and description of the (**a**) force plate and (**b**) foot accelerometer gait event detection methods.

**Figure 2 sensors-19-01483-f002:**
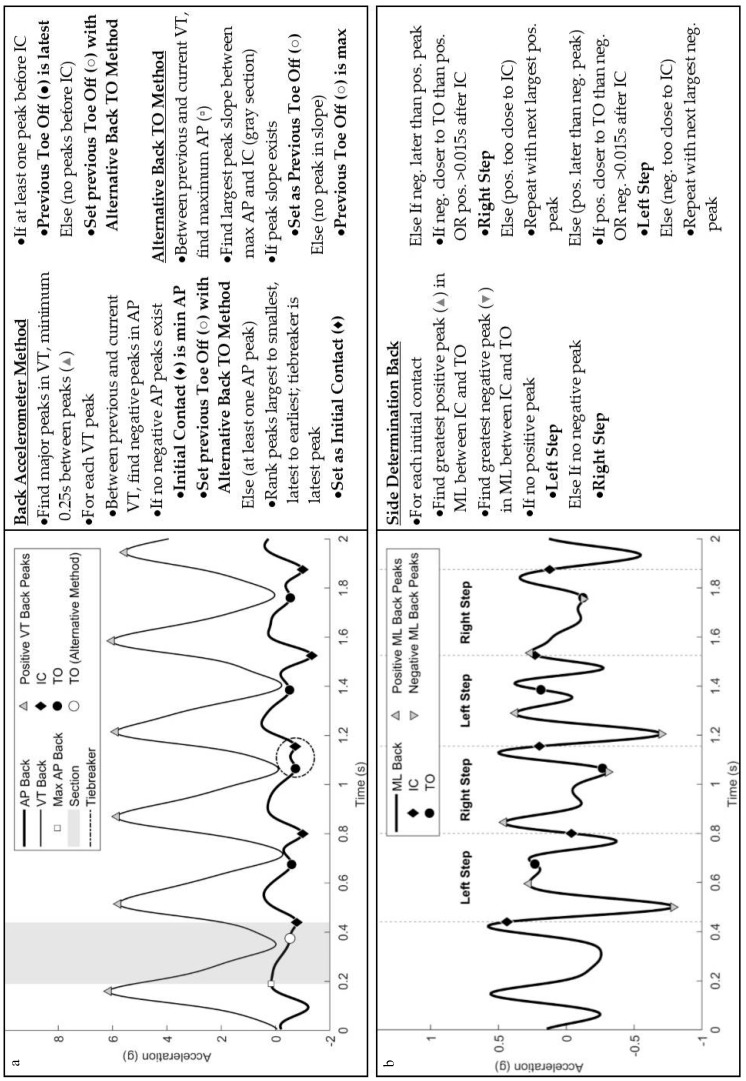
Illustration and description of the back accelerometer (**a**) gait event detection method and (**b**) determination of left or right-side steps.

**Figure 3 sensors-19-01483-f003:**
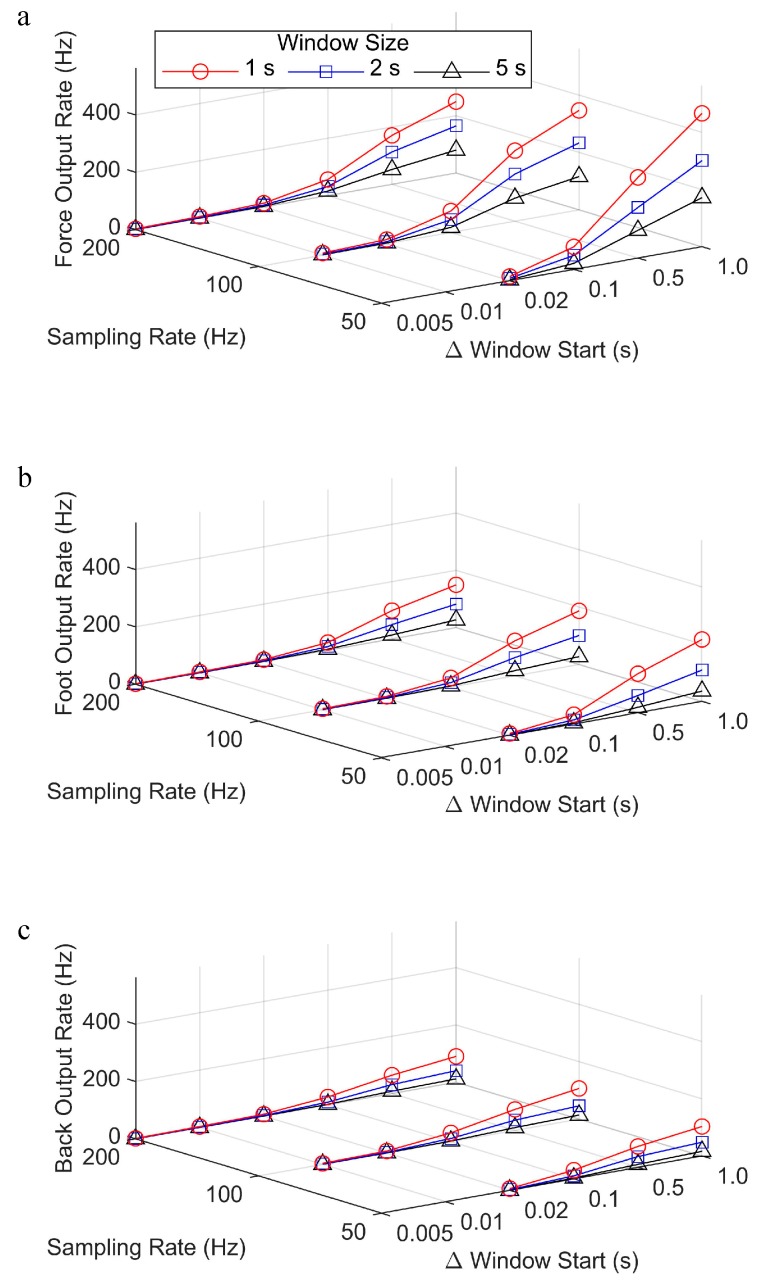
Output rate for the (**a**) force, (**b**) foot, and (**c**) back gait event detection methods, processed with different sampling rate, window size, and change in window start time parameters.

**Figure 4 sensors-19-01483-f004:**
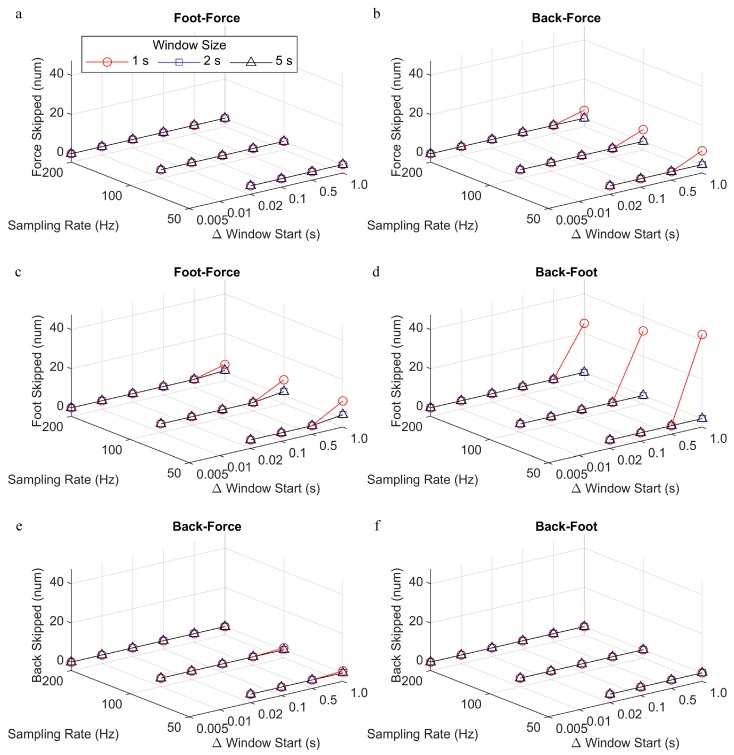
Number of total skipped steps (out of a mean of 1942 recorded steps) for the force (top row), foot (middle row) and back (bottom row) gait event detection methods and each comparison (foot-force, back-force, and back-foot), processed with different sampling rate, window size, and change in window start time parameters. The number of skipped steps is reported for (**a**) force method, foot-force comparison; (**b**) force method, back-force comparison; (**c**) foot method, foot-force comparison; (**d**) foot method, back-foot comparison; (**e**) back method, back-force comparison; (**f**) back method, back-foot comparison. Note, only right-side steps were compared in the foot-force and back-foot comparisons.

**Figure 5 sensors-19-01483-f005:**
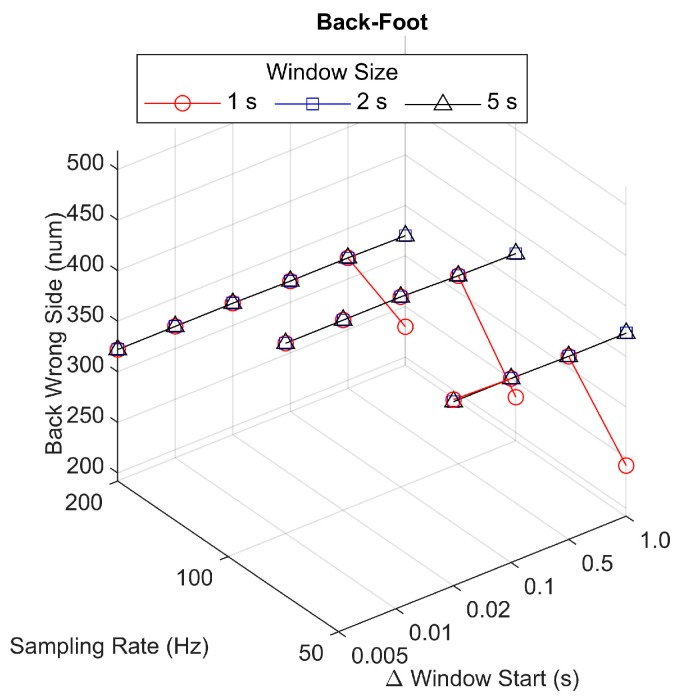
Number of steps (out of a mean of 1942 recorded steps) in which the back accelerometer method assigned the wrong step among different sampling rate, window size, and change in window start time parameters. The number of incorrect step side assignments was determined based on the back-foot comparison.

**Figure 6 sensors-19-01483-f006:**
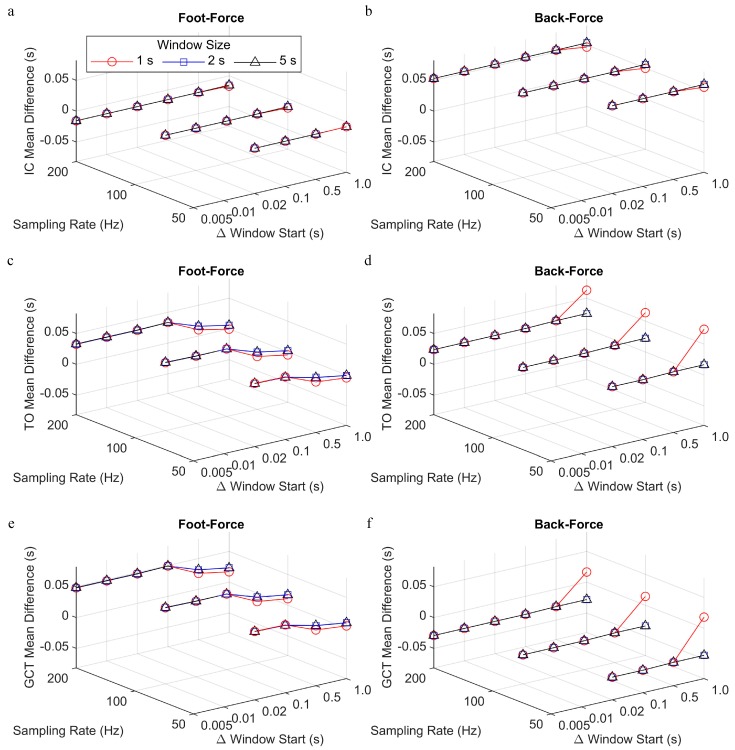
Mean difference between the accelerometer-based and force plate-based identification of initial contact (IC), toe off (TO), and ground contact time (GCT), processed with different sampling rate, window size, and change in window start time parameters. The difference is reported for (**a**) IC, foot-force comparison; (**b**) IC, back-force comparison; (**c**) TO, foot-force comparison; (**d**) TO, back-force comparison; (**e**) GCT, foot-force comparison; (**f**) GCT, back-force comparison.

**Figure 7 sensors-19-01483-f007:**
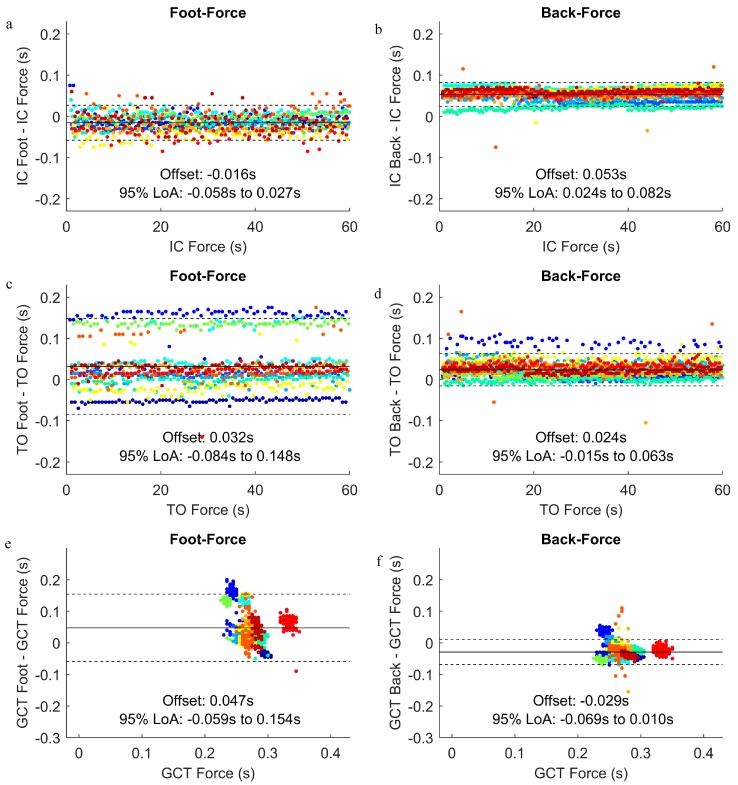
Bland-Altman plots and 95% limits of agreement between the accelerometer-based and force plate-based identification of IC, TO, and GCT over the course of a 60 s running trial. Each dot represents one gait event (IC of TO) or step (GCT), and each participant is plotted with a unique color. The data were processed using a sampling rate of 200 Hz, window size of 2 s, and change in window start time of 0.02 s. The plots are provided for (**a**) IC, foot-force comparison; (**b**) IC, back-force comparison; (**c**) TO, foot-force comparison; (**d**) TO, back-force comparison; (**e**) GCT, foot-force comparison; (**f**) GCT, back-force comparison.

**Figure 8 sensors-19-01483-f008:**
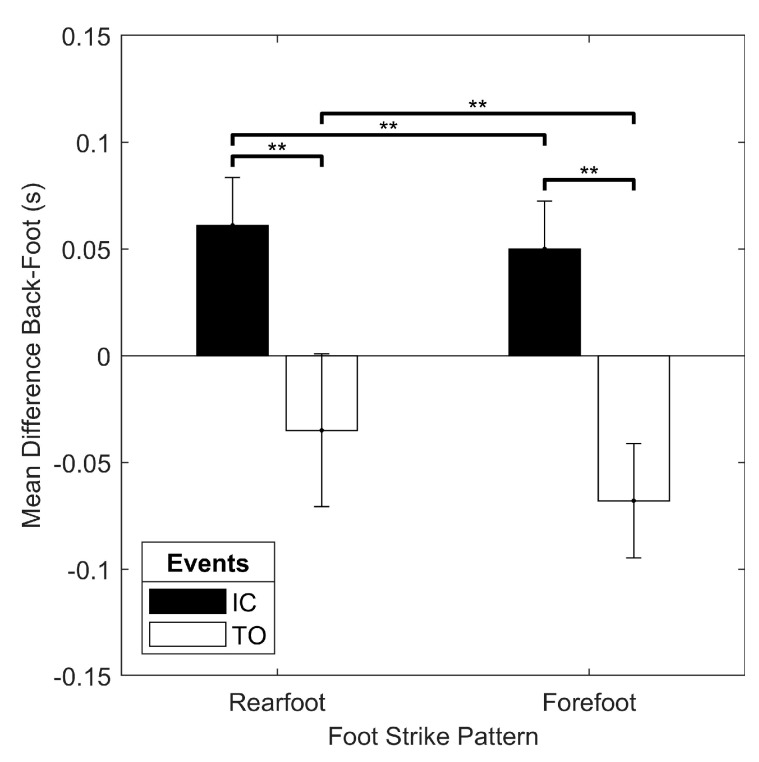
Results of the foot strike x event interaction on the mean difference between the back and foot algorithms for the indoor track running conditions. ** simple main effect significant at *p* < 0.01.

**Figure 9 sensors-19-01483-f009:**
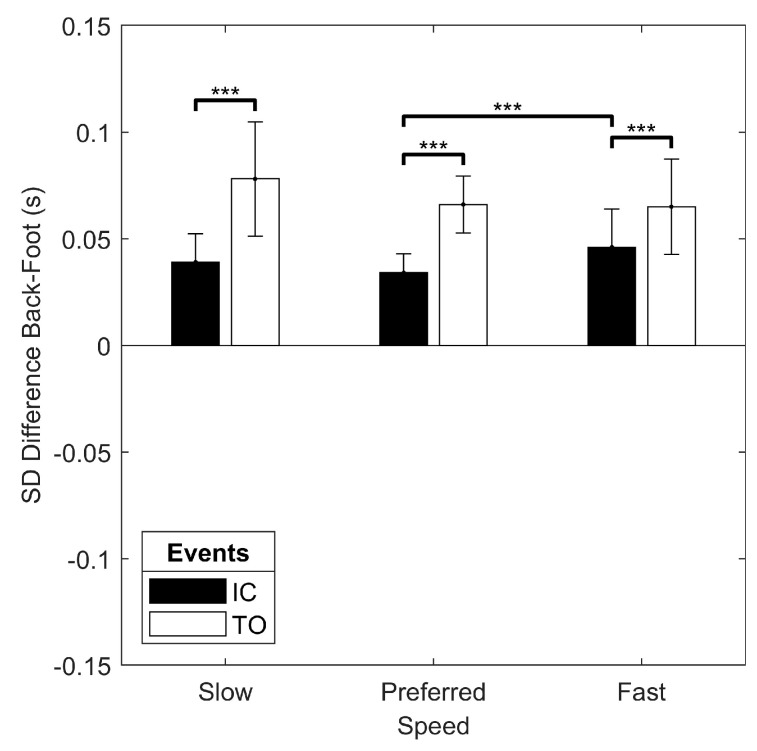
Results of the speed x event interaction on the standard deviation of the difference between the back and foot algorithms for the indoor track running conditions. *** simple main effect significant at *p* < 0.001.

**Figure 10 sensors-19-01483-f010:**
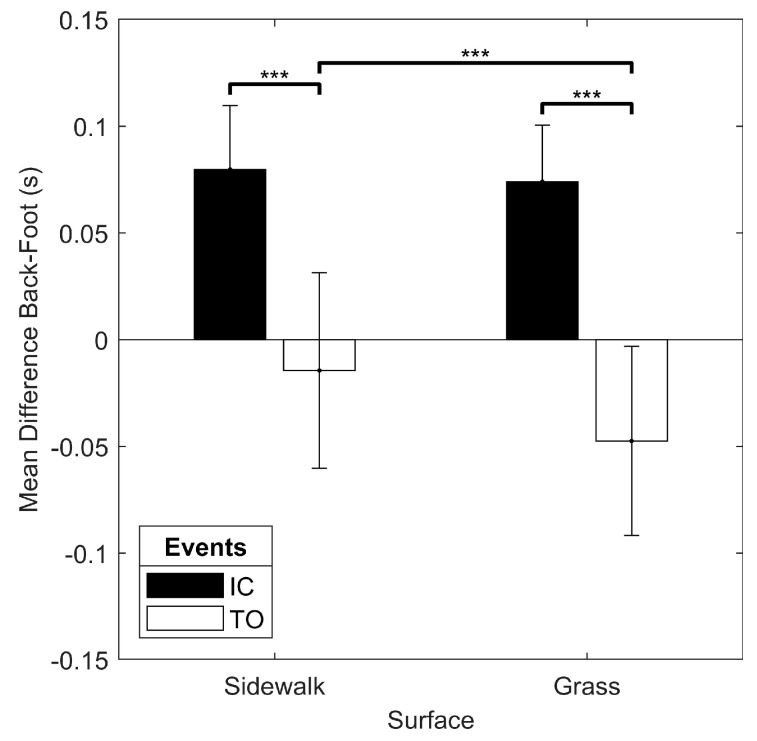
Results of the surface x event interaction on the mean difference between the back and foot algorithms for the outdoor running conditions. *** simple main effect significant at *p* < 0.001.

**Table 1 sensors-19-01483-t001:** The mean and standard deviation (SD) of the back-foot difference in all running conditions.

Experiment	Condition	Mean Back-Foot Difference (s)		SD Back-Foot Difference (s)	
		IC	TO	IC	TO
Treadmill	2.7 m/s	0.066 (0.013)	−0.015 (0.036)	0.018 (0.010)	0.041 (0.027)
	3.3 m/s	0.069 (0.020)	−0.008 (0.053)	0.014 (0.007)	0.026 (0.017)
	3.6 m/s	0.065 (0.017)	−0.009 (0.054)	0.014 (0.006)	0.023 (0.018)
Indoor Track	Rearfoot, Slow	0.057 (0.024)	−0.030 (0.058)	0.049 (0.025)	0.088 (0.032)
	Rearfoot, Preferred	0.066 (0.025)	−0.029 (0.041)	0.039 (0.015)	0.069 (0.022)
	Rearfoot, Fast	0.059 (0.019)	−0.046 (0.030)	0.049 (0.021)	0.069 (0.017)
	Forefoot, Slow	0.047 (0.024)	−0.066 (0.052)	0.030 (0.012)	0.067 (0.032)
	Forefoot, Preferred	0.051 (0.026)	−0.058 (0.042)	0.029 (0.011)	0.063 (0.017)
	Forefoot, Fast	0.052 (0.019)	−0.079 (0.026)	0.042 (0.015)	0.061 (0.026)
Outdoor	Sidewalk	0.080 (0.030)	−0.015 (0.046)	0.032 (0.019)	0.067 (0.019)
	Grass	0.074 (0.027)	−0.048 (0.044)	0.033 (0.023)	0.064 (0.025)

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
