# Peer review of "Automated Accelerometer-Based Gait Event Detection During Multiple Running Conditions"

_sensors, 2019, doi:10.3390/s19071483_

Reviewer 1 Report

In the paper the authors proposed one method for initial contact (IC) and toe off (TO) events detection for gait analysis. The data is collected from accelerometers mounted on foot and back of runners. 

The authors should explain the potential usage of the proposed method. In the paper, it is just mentioned that 'running gait analyses can be used for injury prevention and treatment as well as performance enhancement'. There is not any other content on its usages and applications. 

The proposed methods describe in Figure 1 and Figure 2 are just the combination of some hand-crafted rules. Is it possible to design some automatic learning methods for the IC and TO detection? If they are just the combination of hand-crafted rules, how to ensure that the parameters used in the methods are the best? Is there solution to optimize them?

The titles for Section 2.4 and Section 3.1 are the same 'Algorithm Automation'. What 'Algorithm Automation' means? I was confused by it.

I think the paper is like one technical report, not an research paper. 

Author Response

Reviewer 1

In the paper the authors proposed one method for initial contact (IC) and toe off (TO) events detection for gait analysis. The data is collected from accelerometers mounted on foot and back of runners.

The authors should explain the potential usage of the proposed method. In the paper, it is just mentioned that 'running gait analyses can be used for injury prevention and treatment as well as performance enhancement'. There is not any other content on its usages and applications.

Thank you for this suggestion.  We have added some more information regarding the type of variables that can be computed from accelerometer signals but require the identification of gait events (Line 32-34), and how these variables may be useful in injury prevention (Line 35-37).  Additionally, we stated the goal of the paper, namely that use of these algorithms will further our understanding of real-world running biomechanics (Line 74-77).

The proposed methods describe in Figure 1 and Figure 2 are just the combination of some hand-crafted rules. Is it possible to design some automatic learning methods for the IC and TO detection? If they are just the combination of hand-crafted rules, how to ensure that the parameters used in the methods are the best? Is there solution to optimize them?

It is a good point that a machine learning algorithm may identify IC and TO instead of the specific steps outlined in this study.  However, the algorithms in this study are based on previously described patterns in running accelerometer signals, and have a biomechanical basis.  We have included this reasoning (Line 158-162).

The titles for Section 2.4 and Section 3.1 are the same 'Algorithm Automation'. What 'Algorithm Automation' means? I was confused by it.

We have changed these titles to “Algorithm Implementation” as they include the steps needed to implement the algorithm to process large quantities of data (Line 172-173, 258).

I think the paper is like one technical report, not an research paper.

We agree that this paper is very technical, however, several experiments were conducted, and we believe that this paper fits within the aims and scope of this journal, particularly this special edition of Wearable Sensors for Gait and Motion Analysis.

Reviewer 2 Report

The article describes gait event detection algorithms for running. This is an interesting paper but there are a minor of issues that would have to be addressed beforehand.

  1. Introduction needs to be rewritten. It is not clear what the importance of this objective. Motivation is not clear, need to strengthen.
    2. Materials and methods:
    - Please add information about subjects' BMI.
    - Please clarify subject selection criteria.
    - Did the ground recation force from a force plate was normalized to the subject's body mass?
    3. I would advise recognized the limitation of including a  small group of subjects in the Experiment 1.

Author Response

Reviewer 2

The article describes gait event detection algorithms for running. This is an interesting paper but there are a minor of issues that would have to be addressed beforehand.

1. Introduction needs to be rewritten. It is not clear what the importance of this objective. Motivation is not clear, need to strengthen.
Thank you for this suggestion.  We have added some more information regarding the type of variables that can be computed from accelerometer signals but require the identification of gait events (Line 32-34), and how these variables may be useful in injury prevention (Line 35-37).  Additionally, we stated the goal of the paper, namely that use of these algorithms will further our understanding of real-world running biomechanics (Line 74-77).

2. Materials and methods:

- Please add information about subjects' BMI.

We appreciate this suggestion, however, we have provided the height and mass for the participants, which could be used to calculate BMI if a reader desired this information.  We don’t feel this additional information is relevant to this particular study, but the information is available if a reader would like to make this calculation.

- Please clarify subject selection criteria.

Thank you for this suggestion.  We have added subject selection criteria (Line 82-83).

- Did the ground recation force from a force plate was normalized to the subject's body mass?

Thank you for this question.  Consistent with the previous study referenced for the data processing, there was no normalization of the ground reaction force or accelerometer signals, and this is now explicitly stated in the manuscript (Line 135).

3. I would advise recognized the limitation of including a  small group of subjects in the Experiment 1.

Thank you for this suggestion.  We have included this as a potential direction for future studies (Line 444-445).

Reviewer 3 Report

This study proposed new algorithms which may detect gait event automatically from accelerometer data obtained from multiple running conditions. The novelties of this study are the ability of the algorithms to detect gait event automatically based on the signal pattern from accelerometer and the reliability of the algorithms to conduct any real-time evaluation of running patterns in real word running condition without any expensive equipment such as 3D motion capture and force plate which are limited in laboratory scale experiment. Overall, the proposed algorithm has shown good results to help researcher to evaluate gait events and running during real-life running conditions. However, there are some key points which need to be explained in this study since they, in my opinion, are crucial for further evaluation.

Some detail comments for this study are shown below:

1.       The sampling rate of the accelerometer used in this study need to be reported when describing the Spec of the sensor.

2.       The percentage of overlapping used in this study for the sliding window approach need to be reported when describing the analysis method.

3.       The amount of windows used for each trial and each sliding windows parameter should be reported when describing the analysis method.

4.       Details description regarding to the analysis, such as relationship between the number of windows and the sliding windows parameter, may help to better clarification of the methods.

5.       Why did the author choose the 25% value faster and slower than preferred speed in section 2.1.2 indoor track test? It is suggested to cite reference or supporting information for it.

6.       Why did the author choose 5% strictness for those faster and slower than preferred speed? It is suggested to cite reference or supporting information for it.

7.       Justification is also needed while the mentioned distance value in section 2.1.3 outdoor test.

8.       As comment # 5~6, it is also suggested to justify the reason for choosing the mentioned value of 5% and 15 seconds in section 2.2 data processing.

9.       Citing reference for lines 244 – 246 is suggested.

10.   Did figures 1 & 2 come from or modified from those in references #13 and #18? If yes, to avoid confusion, it is recommended to give citation of these references in the figure caption.

11.   Among the studies being related to pattern in accelerometer signal cited in the main text (refs. 12-21), why did the author choose the references # 13 & 18 as proposed algorithm? Providing proper justification may help the proposed method more persuaded.

Author Response

Reviewer 3

This study proposed new algorithms which may detect gait event automatically from accelerometer data obtained from multiple running conditions. The novelties of this study are the ability of the algorithms to detect gait event automatically based on the signal pattern from accelerometer and the reliability of the algorithms to conduct any real-time evaluation of running patterns in real word running condition without any expensive equipment such as 3D motion capture and force plate which are limited in laboratory scale experiment. Overall, the proposed algorithm has shown good results to help researcher to evaluate gait events and running during real-life running conditions. However, there are some key points which need to be explained in this study since they, in my opinion, are crucial for further evaluation.

Some detail comments for this study are shown below:

1.       The sampling rate of the accelerometer used in this study need to be reported when describing the Spec of the sensor.

Thank you for pointing this out.  We have now included the sampling rate (Line 87)

2.       The percentage of overlapping used in this study for the sliding window approach need to be reported when describing the analysis method.

3.       The amount of windows used for each trial and each sliding windows parameter should be reported when describing the analysis method.

4.       Details description regarding to the analysis, such as relationship between the number of windows and the sliding windows parameter, may help to better clarification of the methods.

To answer points 2, 3, and 4 together, we tested all combinations of multiple sampling rates, window sizes, and changes in window start time (window overlap).  These parameter sets are outlined in section 2.4 (Line 225-227).  During the Algorithm Implementation test, we identified the optimal parameters, which were then used in all future testing of the algorithms.  These optimal parameters are reported in the results (Line 322-324).

5.       Why did the author choose the 25% value faster and slower than preferred speed in section 2.1.2 indoor track test? It is suggested to cite reference or supporting information for it.

Thank you for this suggestion.  We chose 25% as it is a common change from preferred speed in other running biomechanics studies.  Some references are included (Line 110).

6.       Why did the author choose 5% strictness for those faster and slower than preferred speed? It is suggested to cite reference or supporting information for it.

Thank you for this suggestion.  A range of +/- 5% is common among biomechanics studies to ensure participants run close to the prescribed speed.  Some references are included (Line 112).

7.       Justification is also needed while the mentioned distance value in section 2.1.3 outdoor test.

Thank you for pointing this out.  The design of the outdoor run was to mimic real-world running conditions, so we wanted to include a longer running distance, while also ensuring that participants only ran on sidewalk during the sidewalk condition and grass during the grass condition. We have modified the manuscript to emphasize that the sidewalk and grass surfaces were continuous (no mixing of surfaces within conditions) (Line 120-123), and that the result of this outdoor test was nearly 10 km of data for each participant (Line 126-127).

8.       As comment # 5~6, it is also suggested to justify the reason for choosing the mentioned value of 5% and 15 seconds in section 2.2 data processing.

Thank you for this suggestion.  We have provided justification for these values (Line 140-146).

9.       Citing reference for lines 244 – 246 is suggested.

Thank you for this suggestion.  We agree that including the words “by definition” implied that this statement would be defined somewhere when in fact it was intuition that smaller and fewer windows would be processed fastest.  We have modified the sentence to simply report this result (Line 261).

10.   Did figures 1 & 2 come from or modified from those in references #13 and #18? If yes, to avoid confusion, it is recommended to give citation of these references in the figure caption.

This is a good question, but no, these are original figures.

11.   Among the studies being related to pattern in accelerometer signal cited in the main text (refs. 12-21), why did the author choose the references # 13 & 18 as proposed algorithm? Providing proper justification may help the proposed method more persuaded.

Thank you for this suggestion.  We have added our justification for utilizing these references as the basis for our algorithm (Line 158-162).

Round  2

Reviewer 1 Report

The manuscript has been improved in this revision. The authors can reorganize the content of the paper. The authors can introduce the sensors and the data format in Section 2 followed by the algorithm description. And then the experimental data and design is introduced in Section 4. The experimental results and analysis can be presented in Section 5. Section 5, conclusions. Those are just my suggestions. The authors can adjust to the best organization.

Could the authors explain why to choose the results of the method in [20] as the gold standard?

 What is the meaning of the title of Section 2.3 'Algorithm development'? Section 2.4 and Section 3.1, Section 2.5 and Section 3.2 have the same title. Why?

 Page 16: The words between-subject and within-subject can be inter-subject and intra-subject.

 The evaluation standard (mean, difference, etc...) can be described in a subsection to make it clearer.

 The description of the proposed method is not straightforward. I suggestion to rewrite to make it clearer.

Author Response

The manuscript has been improved in this revision. The authors can reorganize the content of the paper. The authors can introduce the sensors and the data format in Section 2 followed by the algorithm description. And then the experimental data and design is introduced in Section 4. The experimental results and analysis can be presented in Section 5. Section 5, conclusions. Those are just my suggestions. The authors can adjust to the best organization.

Thank you for these suggestions.  We agree that this may be a reasonable way to outline this paper.  However, we have chosen the current structure based on the journal’s Instructions for Authors, which state that the Research Manuscript Sections should be: Introduction, Materials and Methods, Results, Discussion.  We have included sub-headings in the Materials and Methods section and the Results section to further organize the manuscript as you have suggested.  Specifically, when we have used the same sub-headings in the Materials and Methods section and the Results section, it is to indicate that the methods and results in these sections are complementary. 

Could the authors explain why to choose the results of the method in [20] as the gold standard?

Thank you for pointing this out.  We have revised this sentence to indicate that ground reaction force thresholds from a force plate are considered the gold standard, and that the study by Mo and Chow utilized this gold standard when validating accelerometer-based methods of gait event detection (Line 51-53).

What is the meaning of the title of Section 2.3 'Algorithm development'? Section 2.4 and Section 3.1, Section 2.5 and Section 3.2 have the same title. Why?

In Section 2.3, we mean to describe the specific algorithm that was developed for gait event detection in this study.  To make this more clear, we have changed this sub-heading to Algorithm Description (Line 148).  As explained in the answer to your first question, when we have used the same sub-headings in the Materials and Methods section and the Results section, it is to indicate that the methods and results in these sections are complementary (i.e. for the methods described in Section 2.4, the results are found in Section 3.1). 

Page 16: The words between-subject and within-subject can be inter-subject and intra-subject.

We agree that the terms inter-subject and intra-subject could be used.  However, since they are very similar in spelling, it may be easy to confuse them when both are used in the same sentence.  Therefore, we prefer to use between-subject and within-subject, which are also acceptable terms.

The evaluation standard (mean, difference, etc...) can be described in a subsection to make it clearer.

Thank you for this suggestion.  For each of the differences reported (foot-force, back-force, back-foot) we have indicated the specific direction of the difference (foot minus force, back minus force, back minus foot).  We hope this clarifies how differences between methods were established (Line 222-223, 244).

The description of the proposed method is not straightforward. I suggestion to rewrite to make it clearer.

Thank you for this suggestion.  We believe the best way to describe the proposed method is to show it in a figure with a step-by-step description of the algorithm, which is done in Figure 1 and 2.  Within the text, we have indicated that these detailed steps are shown in the figures (Line 165).

Reviewer 3 Report

Thank you for the authors' reply and providing necessary additional references which will help the reader in other field understand more. The authors have answered all the comments with required modification in the text.

Author Response

Thank you for the authors' reply and providing necessary additional references which will help the reader in other field understand more. The authors have answered all the comments with required modification in the text.

Thank you for your contributions to this manuscript!

Round  3

Reviewer 1 Report

The manuscript has been improved in this revision. All questions have been addressed. I have no more questions and comments on the paper.